# Redundancy Exploitation of an 8-DoF Robotic Assistant for Doppler Sonography

**Elie Gautreau** [1], **Juan Sandoval** [1,*], **Aurélien Thomas** [1], **Jean-Michel Guilhem** [2], **Giuseppe Carbone** [3,*], **Saïd Zeghloul** [1] and **Med Amine Laribi** [1]

1   Department of GMSCP, Prime Institute CNRS, ENSMA, University of Poitiers, 86073 Poitiers, France;
    elie.gautreau@univ-poitiers.fr (E.G.); thomas.aurelien.perso@gmail.com (A.T.);
    said.zeghloul@univ-poitiers.fr (S.Z.); med.amine.laribi@univ-poitiers.fr (M.A.L.)
2   Independent Researcher, 4 Rue de Coumasaout, 31000 Toulouse, France; jean-michel.guilhem@orange.fr
3   Department of Mechanical, Energy and Management Engineering, University of Calabria, 87036 Rende, Italy
*   Correspondence: juan.sandoval@univ-poitiers.fr (J.S.); giuseppe.carbone@unical.it (G.C.)

**Abstract:** The design of a teleoperated 8-DoF redundant robot for Doppler sonography is detailed in this paper. The proposed robot is composed of a 7-DoF robotic arm mounted on a 1-DoF linear axis. This solution has been conceived to allow Doppler ultrasound examination of the entire patient's body. This paper details the design of the platform and proposes two alternative control modes to deal with its redundancy at the torque level. The first control mode considers the robot as a full 8-DoF kinematics chain, synchronizing the action of the eight joints and improving the global robot manipulability. The second control mode decouples the 7-DoF arm and the linear axis controllers and proposes a switching strategy to activate the linear axis motion when the robot arm approaches the workspace limits. Moreover, a new adaptive Joint-Limit Avoidance (JLA) strategy is proposed with the aim of exploiting the redundancy of the 7-DoF anthropomorphic arm. Unlike classical JLA approaches, a weighting matrix is actively adapted to prioritize those joints that are approaching the mechanical limits. Simulations and experimental results are presented to verify the effectiveness of the proposed control modes.

**Keywords:** medical robot; redundancy resolution; human-robot interaction; torque-control; Doppler sonography

## 1. Introduction

Nowadays, the use of medical robot assistants arises as a suitable solution to improve the working conditions of practitioners, cooperating with them to accomplish the medical tasks. Generally, the quality of the executed task is improved through this collaboration, where the medical capabilities of the expert are magnified by the robot, resulting in improved precision in tasks and lower time of execution while guaranteeing the well-being of the practitioner [1]. Thereby, several medical robot assistants are currently used in the operating room, such as the da Vinci Surgical System [2] from Intuitive Surgical, which has been the market leader for years. Similar examples can be found in other surgical specialties such as neurological and spine surgery [3], joint replacement surgery [4] or laparoscopic surgery [5].

Non-invasive applications can significantly benefit from medical robotic assistants. This is the case in Doppler sonography application, where a number of studies have recently been conducted to propose efficient robotic assistants aiding the specialists to improve their working conditions [6–8]. Indeed, the practitioner must adopt uncomfortable postures during the execution of standard ultrasound examinations. This often makes sonographers suffer musculoskeletal disorders early in their careers. To address this issue, several teleoperated robotic solutions have been proposed in recent years, mostly using commercial robotic arms as a probe-holder [9–11]. We have also recently proposed a teleoperated robotic

assistant using a 7-DoF anthropomorphic arm as probe-holder [7,8]. The practitioner controls the robot by handling a haptic interface into a comfortable workspace. The main drawback of these proposed systems is the limited robot workspace, excluding the possibility of realizing an exam in the whole patient's body. Mobile solutions, such as the one proposed in [12], overcome this problem but need the aid of a human assistant to hold the mobile robot over the patient. A new version of the system proposed in [8] has recently been introduced in [13], including a motorized linear axis at the base of the robot to enlarge its workspace and allow a complete exam through the entire patient's body without needing manual readjustments of the platform's position. Knowing that the new platform at the patient site has 8-DoF, the degree of redundancy can be exploited in several ways, for instance, to consider a kinematic constraint [14], to avoid collisions [15] or mechanical joint limits [16], or to optimize a performance criterion such as the manipulability index [17]. This paper proposes two control modes to deal with the redundancy resolution of the 8-DoF robot. Knowing that safe human-robot interaction must be guaranteed, these control modes are proposed at the torque level, in order to implement a compliant behavior of the robot.

The first control mode considers the robot as a full 8-DoF kinematics chain. The advantage of this approach is that the eight joints are simultaneously activated, avoiding the presence of certain singularities typically linked to the limits of the workspace when only moving the 7-DoF arm. Therefore, the manipulability of the platform is naturally improved. The second control mode considers the two systems, the 7-DoF arm and the linear axis, separately. The robotic arm is activated in priority whereas the linear axis is only activated once the arm reaches the desired workspace limits. Furthermore, we present a new adaptive Joint-Limit Avoidance (JLA) strategy with the aim of exploiting the redundancy of the 7-DoF anthropomorphic arm. Classical JLA approaches define the diagonal weighting matrix as constant, which lets a continuous generation of null space torques to avoid joint limits, even for the joints that are far from their limits [16]. An improved approach is proposed here, where the weighting matrix is actively adapted to prioritize those joints approaching the mechanical limits. This feature allows an enhanced distribution of the redundant space of the robot. Moreover, when none of the joints is in the vicinity of the limits, zero null space torque is generated, allowing a free motion of the robot's elbow in case the expert wants to manually reconfigure the robot.

The main contributions of the paper are summarized as follows: (a) detailed presentation of an 8-DoF teleoperated platform for Doppler sonography, (b) validation of a fully redundant control mode at the torque level for the 8-DoF robotic system and (c) introduction of a new adaptive JLA strategy for redundant robots controlled by torque by means of an optimal variation of the weighting matrix.

The paper is organized as follows: Section 2 details the medical requirements and the description of the proposed robotic assistant platform. Sections 3 and 4 present the first and the second control modes, respectively. The last section concludes the presented work and opens the perspectives of future works.

## 2. Robotic-Assistant Platform

This section presents the proposed teleoperated robotic platform for Doppler Sonography. The platform has been designed based on the medical requirements collected through a study of the medical gesture.

### 2.1. Medical Requirements

Doppler ultrasound is a medical imaging method used by sonographers (i.e., angiologists) to study the cardiovascular system of patients using an ultrasound probe. According to the Society of Diagnostic Medical Sonography, 90% of clinical sonographers have experienced symptoms of work-related musculoskeletal disorders (WRMDs), mainly due to an accumulation of repeated gestures in awkward postures and to the frequent application of downward pressure with the probe [18].

In order to confirm this thesis, a study of the angiologist's gestures has been carried out in [7], and we resume in this section the main obtained results. A Motion Capture (MoCap) system, i.e., Qualisys, was used to record several doppler ultrasound examinations made by a practitioner on real patients and under real conditions. Markers used for MoCap were placed both on the practitioner and on the probe to record the right arm, pelvis and head movements of the practitioner during the performed examinations. Moreover, the probe have been instrumented with a FSR sensor to measure the force applied by the specialist (Figure 1). Each patient has been examined in regions all along the body: carotid, legs, and abdomen. The study highlighted an outing from the expert's joint comfort zone, which can cause WRMSDs with repetitive movements. The maximum measured values of the orientation angles considered correspond to the worst postures. The latter are compared to the comfort zone reference angles defined by ISO 11226, ISO 11228-3 and NF EN 1005-4 norms. An example of head rotation and wrist angles for the carotid exam is given in Figure 2. These values, joint angles for neck twist and wrist joint, prove that the angiologist very often works outside the comfort zone described by the standards.

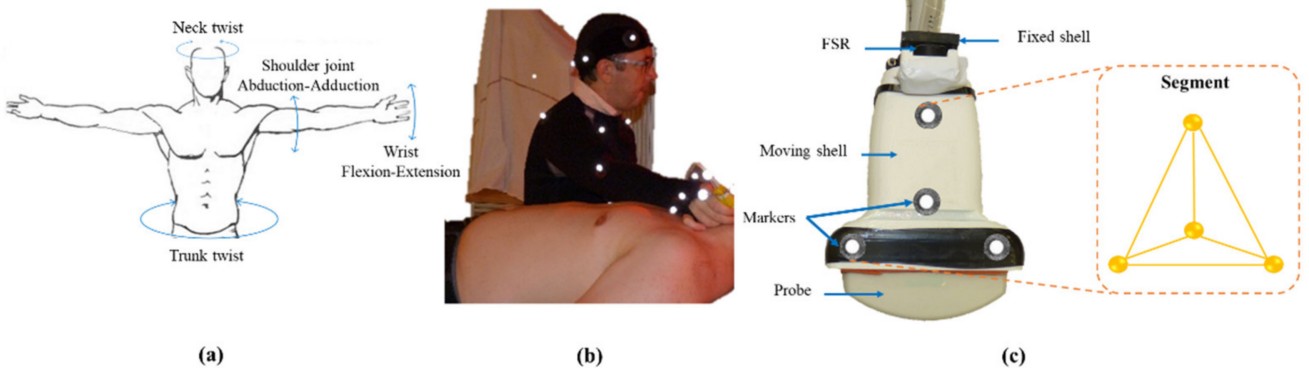

**Figure 1.** Study of the angiologist's gestures using MoCap: (**a**) Measured angles on the angiologist's body. (**b**) Reflective markers fixed to the angiologist's body (**c**) Reflective markers and FSR sensor fixed to the probe.

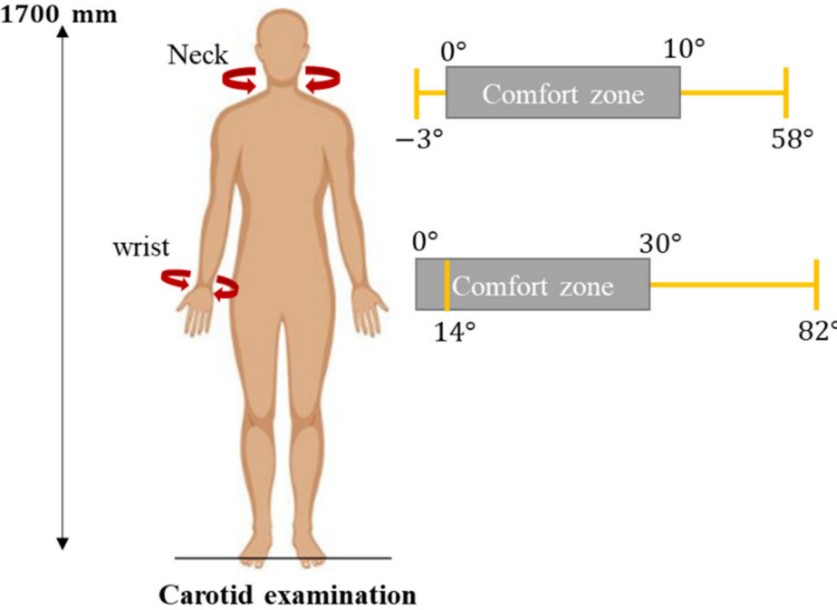

**Figure 2.** Neck and wrist joint range angles measured on angiologist during carotid exams.

In addition, the efforts measured with the instrumented probe revealed that the intensity of the effort depends on the type of examination and the morphology of the

patient and can also lead to the development of MSDs. For instance, the maximum force applied was measured during the examination of an abdomen. A greater effort was required by the angiologist to determine the position of the abdominal aorta, which is particularly recurrent for the category of fat patients.

The interest of this study is twofold. It highlighted (1) the common gestures performed out of the joint comfort zones and (2) the significant efforts to be applied during an examination. The combination of the latter can lead to musculoskeletal disorders during numerous repetitive examinations. In order to overcome these two observations, a robotic teleoperation platform composed of a haptic interface (expert site) and a cobot (patient site) was developed. Thus, the cobot performs the efforts and postures controlled by the doctor through a haptic interface. Since the cobot workspace is not large enough to cover the whole patient's body, it has been mounted on a motorized linear axis, producing an 8-DoF robotic platform.

### 2.2. Experimental Setup

Figure 3 presents the assembled prototype of the proposed teleoperated robotic platform. The platform at the patient site, i.e., Franka Emika and linear axis, is teleoperated by the practitioner through a 6-DoF haptic interface.

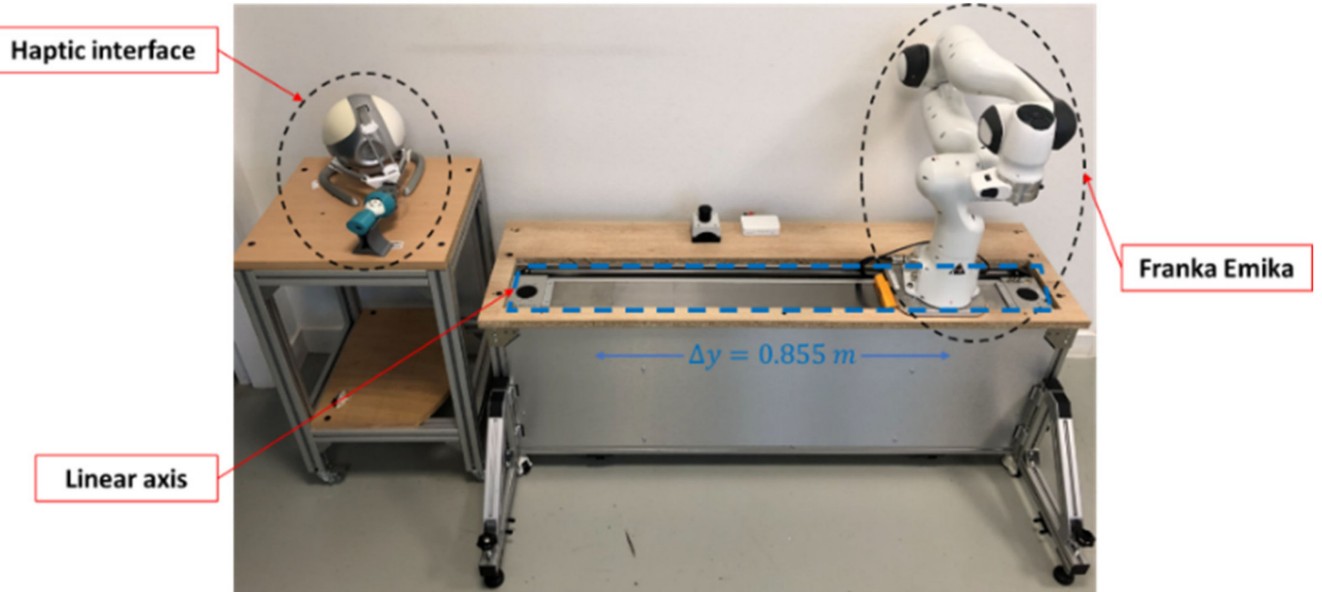

**Figure 3.** Assembled prototype of the 8-DoF robotic platform for medical applications.

The platform's workspace has been increased by 0.855 m on the *Y-axis* thanks to range of motion of the linear axis. This latter is powered by a Kollmorgen brushless motor via a belt drive. The motor driver is controlled through a Telnet protocol. Moreover, the high-level control of the overall platform is implemented in the Robot Operating System (ROS) middleware.

### 2.2.1. Expert Site: 6-DoF Haptic Interface

Haptic devices have been widely employed in many areas such as surgery and craniotomy application [19], rehabilitation and manufacturing. In a general framework, haptic devices are based on mechanical interfaces that link tactile information between a human and the device. Various haptic interface architectures have been developed, with either serial or parallel architectures. While serial devices present several drawbacks as inertia, rigidity and positioning issues, parallel interfaces overcome these drawbacks but present limited workspaces and singular configurations issues. Hybrid haptic interface architec-

tures take advantage of the previous ones. Some devices have been developed commercially, such as the Sigma 7 [20] or the 6-DoF interface proposed by Tsumaki et al. [21].

We have developed a 6-DoF hybrid haptic interface, as shown in Figure 4. It is composed of a 3-DoF Novint Falcon interface combined with an inertial measurement unit (IMU) attached with a spherical wrist. Novint Falcon allows translational motions in 3D space, similar to the one proposed by Tsai [22], into a maximum volume of 10.16 cm along each direction, and a maximum force feedback of around 9 N. A LPMS-B2 inertial measurement unit, located inside the artificial probe measures the rotational motions (3-DoF). The artificial probe mounted on the haptic device simulates a real probe for Doppler sonography examinations. Additionally, the system at the expert site includes a foot pedal to prolong the desired motion sent to the robot along the current direction of motion of the haptic interface.

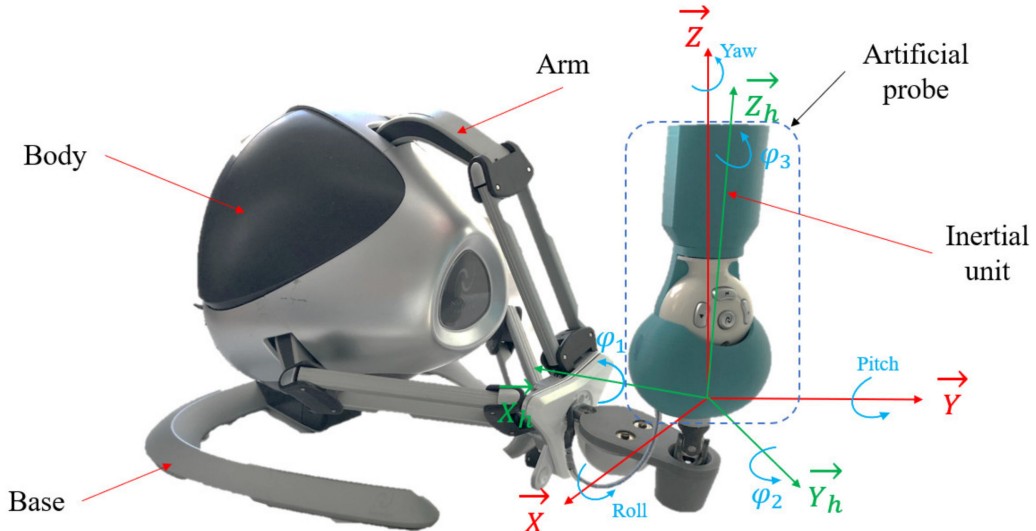

**Figure 4.** 6-DoF hybrid haptic interface.

### 2.2.2. Patient Site: 8-DoF Redundant Robot

As mentioned above, we previously proposed the use of a 7-DoF Franka Emika cobot at the patient site for the robotic assistant platform [7]. Nevertheless, the workspace of the 7-DoF cobot is insufficient to perform an examination of the entire patient's body. Therefore, we propose to enlarge the workspace of the robot by mounting it on a belt-driven linear axis (Parker HMRB15) placed parallel to the longitudinal axis of the patient's table (Figure 3). The linear axis is motorized by a 24 V brushless motor coupled with a gear reducer (12.5 mm/lap). More technical details about the linear axis are presented in Table 1.

**Table 1.** Technical characteristics of the linear axis.

| Axis | Parker HMRB15 |
|---|---|
| Motor | Brushless Motor AKM3 |
| Driver | Kollmorgen AKD-P00306 |
| Maximum Imposed Torque | 3 Nm |
| Linear speed imposed | 10 mm/s |
| Motion amplitude | 900 mm |

When employing the platform, the robotic system is placed next to the patient's table so that the linear axis motion allows the entire table to be covered. It is therefore planned that the patient's waist is aligned with the middle position of the linear axis, facilitating the identification of his positioning by the practitioner.

### 2.2.3. Communication Framework

Robot Operating System is the middleware used in the platform to guarantee real-time data exchange between the different devices. These applications have been developed in Python and/or C++ and launched in ROS Kinetic version under Ubuntu 16.04 with real-time patched kernel, so that a soft real-time operating system is configured.

The diagram in Figure 5 shows the information transmitted between the different systems. The packages Rosfalcon, Joy and Lmps_IMU are, respectively, those of the Novint Falcon, the foot pedal and the inertial measurement unit composing the hybrid haptic interface. Concerning the operator site, the package Franka_ros allows to read/write commands to the cobot's controller and the package Axis communicates with the controller of the linear axis. Finally, the motion of the axis is determined by the Pos_criteria package, according to the control mode implemented (see Sections 3 and 4).

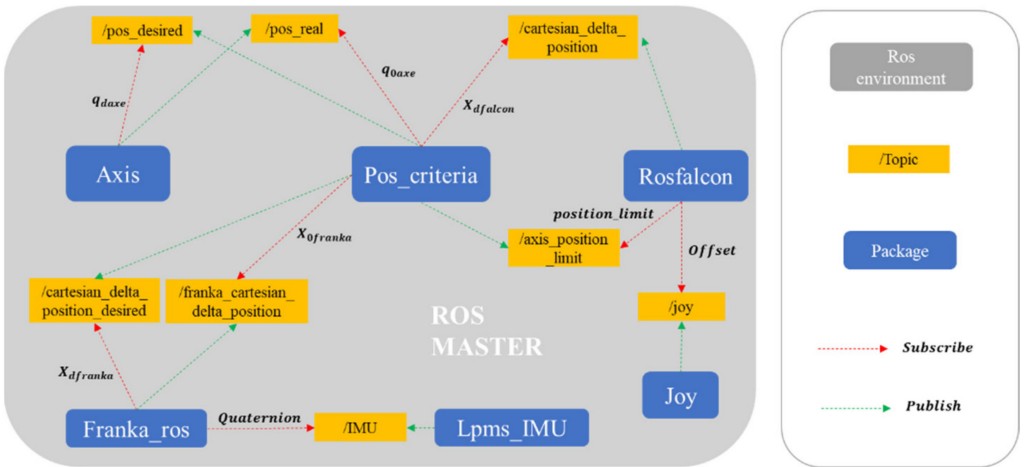

**Figure 5.** Data exchange of the platform in ROS environment.

Communication between the different systems is achieved through Bluetooth (inertial unit), USB (haptic interface), Ethernet (Franka Emika) and Telnet (linear axis) protocols. In the case of the linear axis, telnet protocol has been chosen for its ease of use, as it uses a Telnet library, i.e., Telnetlib in python.

As the linear axis has been implemented as an upgrade, the package Axis has been created and linked to the existing devices. This package enables the control of the linear axis in either position, speed or torque (compliance). The latter gives us the possibility to achieve compliance control. The ROS node to control the linear axis in torque is depicted in Figure 6.

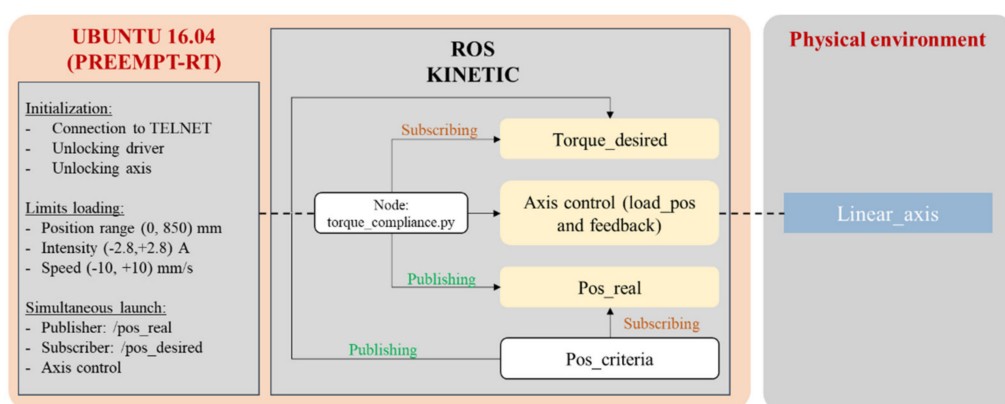

**Figure 6.** Block diagram of the Axis torque node.

The layout of the three nodes corresponding to the position, speed and torque control remains the same. The speed limit stands low as this application aims to deal with patients, thus security is an important issue. Moreover, in the case of high speed, where robot joints can be damaged, the robot will detect this and freeze, preventing the robot from being used.

Two alternative control modes for the platform at the patient site are proposed in the following sections. The first control mode, presented in Section 3, considers the kinematics of the robot as a fully 8-DoF system. Therefore, the motions of the 8 axes are synchronized to fulfill the desired task and the two degrees of redundancy guarantee the avoidance of singularities of the robot arm. This control mode has for now been validated in simulation. The second control mode, presented in Section 4 and validated experimentally, decouples the motion of the linear axis and the robot arm. The linear axis is only activated if the robot arm approaches its workspace limits. Furthermore, the remaining degree of redundancy of the robot arm is employed to avoid the joint limits through a new adaptive JLA strategy.

## 3. Fully Redundant Control Mode

### 3.1. Robot Modelling

As mentioned above, the first control mode considers the robot model of an 8-DoF system. Here below we detail the kinematic and dynamic model of the proposed platform. First, the Denavit–Hartenberg parameters' table [23] of the platform is presented in Figure 7, where joint $i = 0$ corresponds to the linear axis and joints $i = 1, \ldots, 7$ are related to the cobot arm.

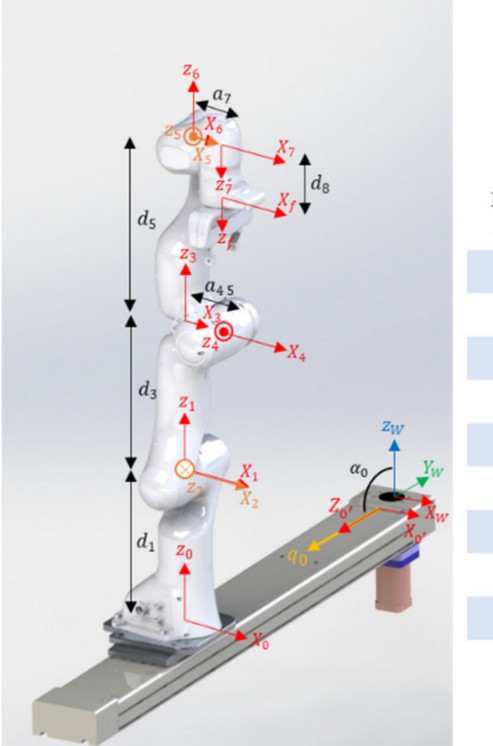

**D-H parameter table**

| Robot joints | Joints $i$ | $a_i$ (m) | $d_i$ (m) | $\alpha_i$ (rad) | $\theta_i$ (rad) |
|---|---|---|---|---|---|
| J0 | 0 | 0 | $q_0$ | $\alpha_0$ | 0 |
| J1 | 1 | 0 | $d_1$ | $-\pi/2$ | $q_1$ |
| J2 | 2 | 0 | 0 | $-\pi/2$ | $q_2$ |
| J3 | 3 | 0 | $d_3$ | $\pi/2$ | $q_3$ |
| J4 | 4 | $a_4$ | 0 | $\pi/2$ | $q_4$ |
| J5 | 5 | $a_5$ | $d_5$ | $-\pi/2$ | $q_5$ |
| J6 | 6 | 0 | 0 | $\pi/2$ | $q_6$ |
| J7 | 7 | $a_7$ | 0 | $\pi/2$ | $q_7$ |
| Flange | | 0 | $d_8$ | 0 | 0 |

**Figure 7.** D-H parameters of the 8-DoF robotic assistant platform.

Considering that a Doppler ultrasound exam requires that the probe fully moves in the space, a task-space dimension of $m = 6$ is defined. Then, let define the non-squared Jacobian matrix $\mathbf{J}(\mathbf{q}) \in \Re^{m \times n}$ with $n = 8$ denoting the number of robot joints. Therefore, the relation between the joint-space velocity $\dot{\mathbf{q}} \in \Re^n$ and the task-space velocity $\dot{\mathbf{x}} \in \Re^m$ yields,

$$\dot{\mathbf{x}} = \mathbf{J}(\mathbf{q}) \cdot \dot{\mathbf{q}} \tag{1}$$

Since the robotic platform is torque-controlled, let us now define its dynamic equation of motion in joint-space as,

$$\mathbf{M}(\mathbf{q})\ddot{\mathbf{q}} + \mathbf{C}(\mathbf{q}, \dot{\mathbf{q}})\dot{\mathbf{q}} + \mathbf{g}(\mathbf{q}) = \boldsymbol{\tau}_c + \boldsymbol{\tau}_{ext} + \boldsymbol{\tau}_f \tag{2}$$

This model depends on the inertial matrix $\mathbf{M}(\mathbf{q}) \in \Re^{n \times n}$, the centrifugal and Coriolis matrix $\mathbf{C}(\mathbf{q}, \dot{\mathbf{q}}) \in \Re^{n \times n}$ and the vector of gravitational torques $\mathbf{g}(\mathbf{q}) \in \Re^n$. Moreover, vectors $\boldsymbol{\tau}_c \in \Re^n$, $\boldsymbol{\tau}_f \in \Re^n$ and $\boldsymbol{\tau}_{ext} \in \Re^n$ represents the output, friction and external torques, respectively.

This section may be divided by subheadings. In the following, we explain the details of the control law defining $\boldsymbol{\tau}_c$, as well as results of the robot's behavior performed in a customized simulator.

*3.2. Control Law*

A compliant behavior of the robotic platform is suitable to reduce the effects of undesired collisions between the robot and the patient. The following control law allows to reproduce the effects of a mechanical damper-spring system at the cartesian space during the execution of a desired trajectory $\mathbf{x}_d \in \Re^m$,

$$\mathbf{F}_{task} = \mathbf{K}_p(\mathbf{x}_d - \mathbf{x}) - \mathbf{K}_d\dot{\mathbf{x}} \tag{3}$$

where the stiffness and damping effects can be adjusted with the diagonal constant matrices $\mathbf{K}_p \in \Re^{m \times m}$ and $\mathbf{K}_d \in \Re^{m \times m}$, respectively [24].

A joint-torque control law implementing the compliant behavior of Equation (3) can be defined as follows,

$$\boldsymbol{\tau}_c = \mathbf{J}^\mathbf{T} \cdot \mathbf{F}_{task} + \boldsymbol{\tau}_\aleph + \boldsymbol{\tau}_{comp} \tag{4}$$

This control law also includes the torque vector $\boldsymbol{\tau}_{comp} \in \Re^n$, compensating the gravitational and dynamic effects, and the torque vector $\boldsymbol{\tau}_\aleph \in \Re^n$, exploiting the redundancy of the robot. As mentioned above, redundancy can be used in several ways according to the specific needs of the application. For instance, a suitable way to exploit the redundancy is to stabilize the internal motion, yielding,

$$\boldsymbol{\tau}_\aleph = \aleph(\mathbf{q})\left[K_{p_{null}}(\mathbf{q_{init}} - \mathbf{q}) - K_{d_{null}}\dot{\mathbf{q}}\right] \tag{5}$$

Joint torques produced by Equation (5) attempt to keep the joint positions as best as possible at the initial joint configuration $\mathbf{q_{init}} \in \Re^n$. The weight of this law for each joint can be tuned with the values of diagonal constant matrices $K_{p_{null}} \in \Re^{n \times n}$ and $K_{d_{null}} \in \Re^{n \times n}$. In order to guarantee that this control law is only performed in the null-space of the robot, avoiding undesired perturbations in the cartesian-space, the torque vector is premultiplied by a null-space projector $\aleph(\mathbf{q}) = \left(\mathbf{I}_{n \times n} - \mathbf{J}^\mathbf{T}\mathbf{J}^+\right)$, defined in terms of the Moore–Penrose pseudoinverse of $\mathbf{J}(\mathbf{q})$, i.e., $\mathbf{J}^+ = \mathbf{J}^\mathbf{T}\left(\mathbf{J}\cdot\mathbf{J}^\mathbf{T}\right)^{-1}$.

*3.3. Performance Analysis*

To achieve a compliance control mode of the robotic platform, a dynamic simulator was developed in Matlab-Simulink, using the Simscape toolbox. For purpose of realistic simulations, the CAD of the real robotic platform was added to the simulator. The dynamic model was calculated by Simulink based on the geometric and inertial parameters of the assembled multi-body system. This new tool allowed us to verify the performance of the robot when controlling it by torque, based in [25]. Figure 8 shows the flowchart for the simulator operating principle.

The control law is an association of three laws:

- Recursive Newton–Euler algorithm: This algorithm has been implemented to calculate the gravitational, centrifugal and Coriolis compensation torques.

- Null-space control law: The use of a null torque vector enables the robot's internal motions stabilization for a given task. Adjusting the values of damping $K_{d_{null}}$ and stiffness $K_{p_{null}}$ allows the choice of priority motion of either the robot or the linear axis. In order to let the robot and the linear axis moving together, the constant matrices have been set to $K_{p_{null}} = diag(100, 7, 4, 4, 5, 4, 3, 4)$ and each value of $K_{d_{null}}$ has been set to $K_{d_{null\ 1,1}} = 26\sqrt{K_{p_{null\ 0,0}}}$ for linear axis and $K_{d_{null\ j,j}} = 0.9\sqrt{K_{p_{null\ j,j}}}$ for the cobot ($j = 1, \ldots, 7$). It is worth mentioning that these values were determined empirically and are highly dependent of the model uncertainties, e.g., friction forces.
- Cartesian compliance law: The required torque is computed to achieve the imposed task at the cartesian space. Stiffness $\mathbf{K}_p$ and damping $\mathbf{K}_d$ matrices have been set to $\mathbf{K}_p = diag(500, 200, 500, 40, 40, 40)$ and $K_{d_{l,l}} = 2.2\sqrt{K_{p_{l,l}}}$. Empirical values of stiffness and damping are consistent with real values allowed by the Franka cobot.

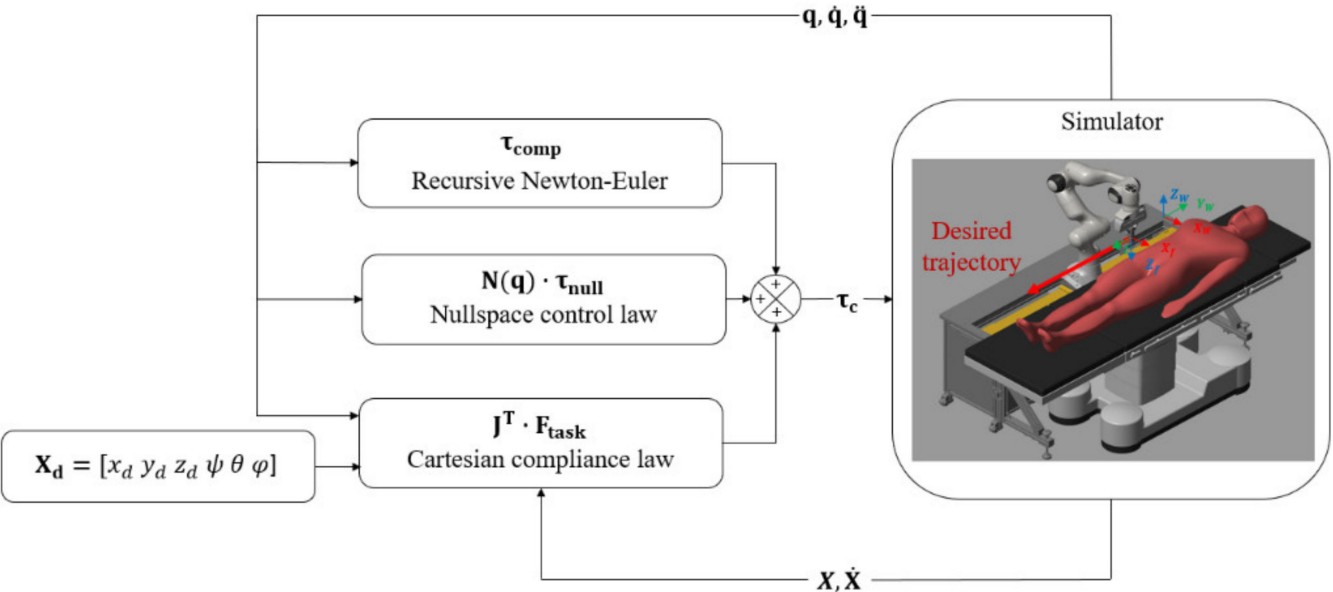

**Figure 8.** Control schema implemented in the dynamic simulator.

A linear trajectory with a total length of 0.8 m along the *Y-axis* has been imposed to validate the correct functioning of the robotic platform when controlled as an 8-DoF redundant robot. The desired linear trajectory has been chosen in such a way that it exceeds the Franka cobot workspace.

In the initial configuration, the robot end-effector cartesian position is set to 0 m along the *Y-axis*. To perform the desired trajectory, a simultaneous motion is conducted by the cobot and the linear axis, as shown in Figures 9 and 10. Indeed, the values of $K_{p_{null}}$ have a high influence on the portion of motion performed by the axis and by the cobot along the *Y-axis*. In this case, we put a larger value to the first diagonal value, related to the action of the linear axis, so that its motion is launched easily. However, in the case that the motion of the linear axis is not a priority, this constant value could be reduced to prioritize the motion of the cobot. A comparison between the desired and the performed trajectories are also presented in Figure 11. It can be seen that the desired trajectory is entirely executed by the platform. Errors between the real and desired motions along the three cartesian axis remain low: $x_{error} = 1.6 \cdot 10^{-5}$, $y_{error} = 1.39 \cdot 10^{-4}$ and $z_{error} = 2.9 \cdot 10^{-5}$, considering that a natural error is induced by the compliance law implemented in Equation (3).

A suitable performance criterion to be analyzed in this new platform is the manipulability index [13], i.e., $\mathbf{M} = \sqrt{\mathbf{J}^T\mathbf{J}}$. It is well-known that the manipulability index decreases when the manipulator approaches a singularity configuration, i.e., $det(\mathbf{J}) = 0$. Naturally, if the linear trajectory is only performed by the cobot arm, the manipulability index would

decrease to zero since the workspace limits would be reached. In contrast, the use of the linear axis and the simultaneous activation of the cobot and the axis allows it to preserve higher manipulability values, such as shown in Figure 12. The manipulability of the robotic platform (8-DoF) is always increasing because of the linear axis's motion. It is also calculated the manipulability of the Franka cobot arm separately, proving that it does not fall, since it is not reaching the limit of its working space.

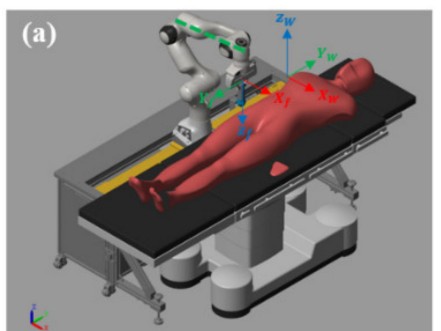
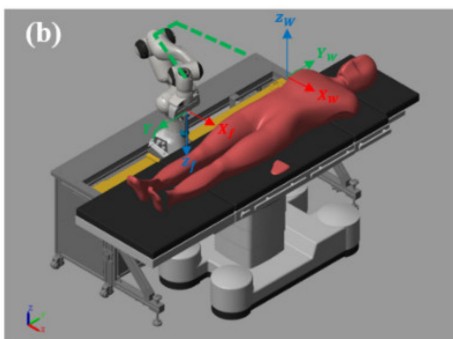
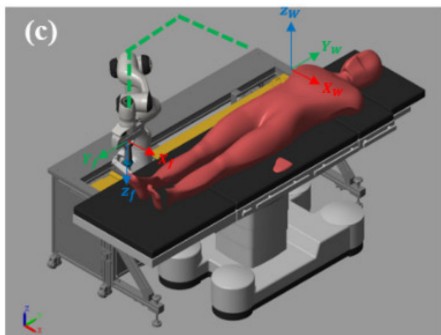

**Figure 9.** Sequential evolution of the robot configuration during the execution of a linear trajectory of 0.8 m along the *Y-axis*. (**a**) Initial, (**b**) intermediate and (**c**) final robot configuration.

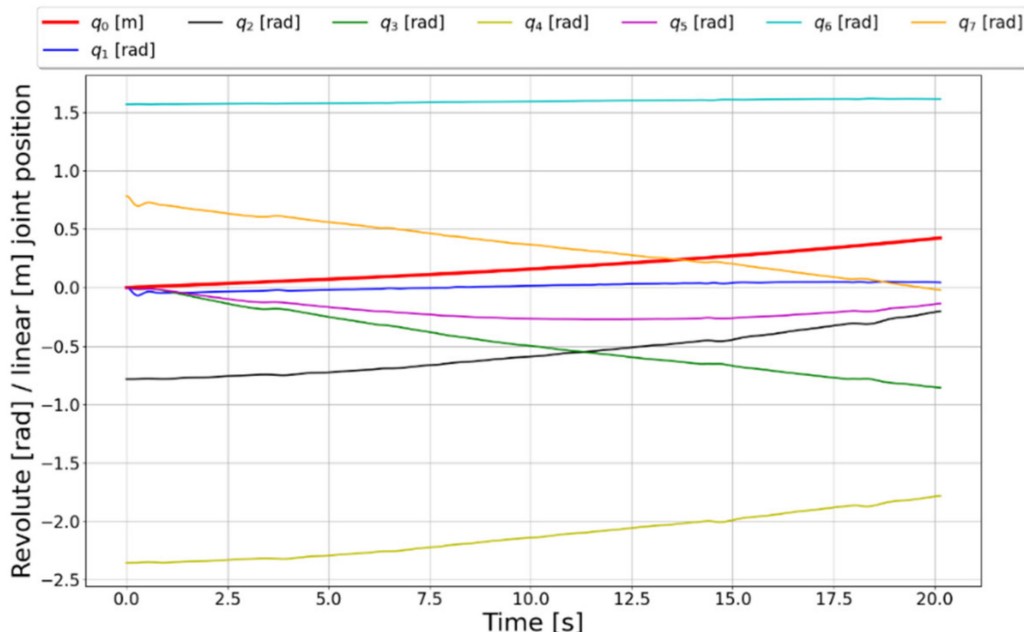

**Figure 10.** Joint position trajectories of the 8-DoF robot assistant during the execution of a linear trajectory along the *Y-axis*.

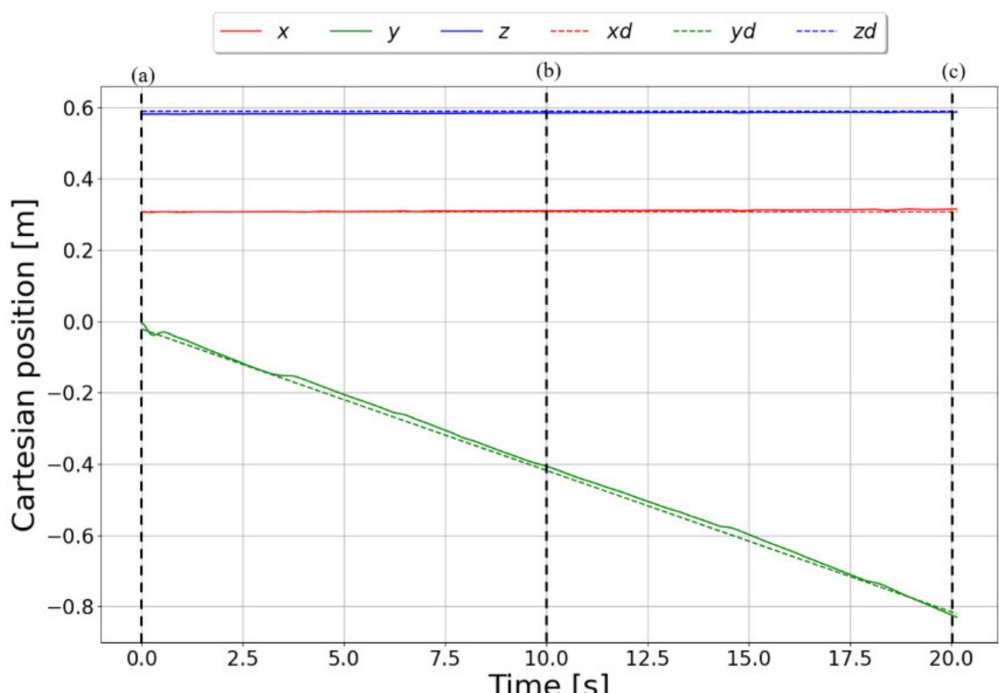

**Figure 11.** Cartesian position trajectory of the 8-DoF robot assistant during the execution of a linear trajectory along the *Y-axis*.

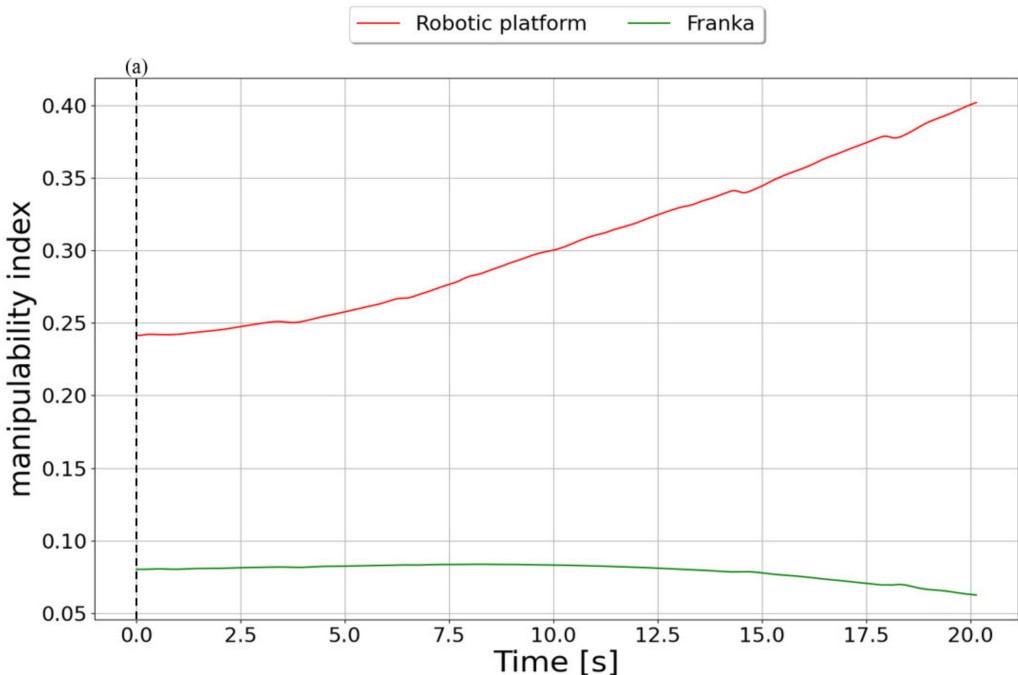

**Figure 12.** Manipulability Index of the 8-DoF robotic platform and the one of the Franka cobot (7-DoF) during the execution of a linear trajectory along the *Y-axis*.

## 4. Decoupled Redundant Control Mode

In this section we propose a decoupled control mode for the proposed 8-DoF redundant robotic platform. This means that both the 7-DoF cobot and the linear axis are controlled independently as two separate systems. The details of this control mode are depicted here below.

### 4.1. Manipulator/Axis Switching Strategy

The control strategy adopted to associate the linear axis and the 7-DoF robot is achieved by decoupling the two devices. The motion distribution between the cobot and the linear axis is achieved through a central ROS node called pos_criteria (Figure 6). A desired cartesian position limit *Xlim* is defined to establish a switching condition determining whether the robot or the linear axis moves along the direction of motion of the linear axis, i.e., *Y-axis*. Upper and lower limits are imposed to $(-0.45, 0.45)$ m. These limits are largely lower than the maximum robot workspace (0.855 m). Two scenarios depending on the position condition are shown in Figure 13.

a.     While the robot's end-effector does not reach the limits, the linear axis remains fixed, and the robot moves along the *Y-axis* to fulfil the desired trajectory.

b.     If the limits are exceeded, the linear axis is launched, and the robot's end-effector remains fixed along the *Y-axis*.

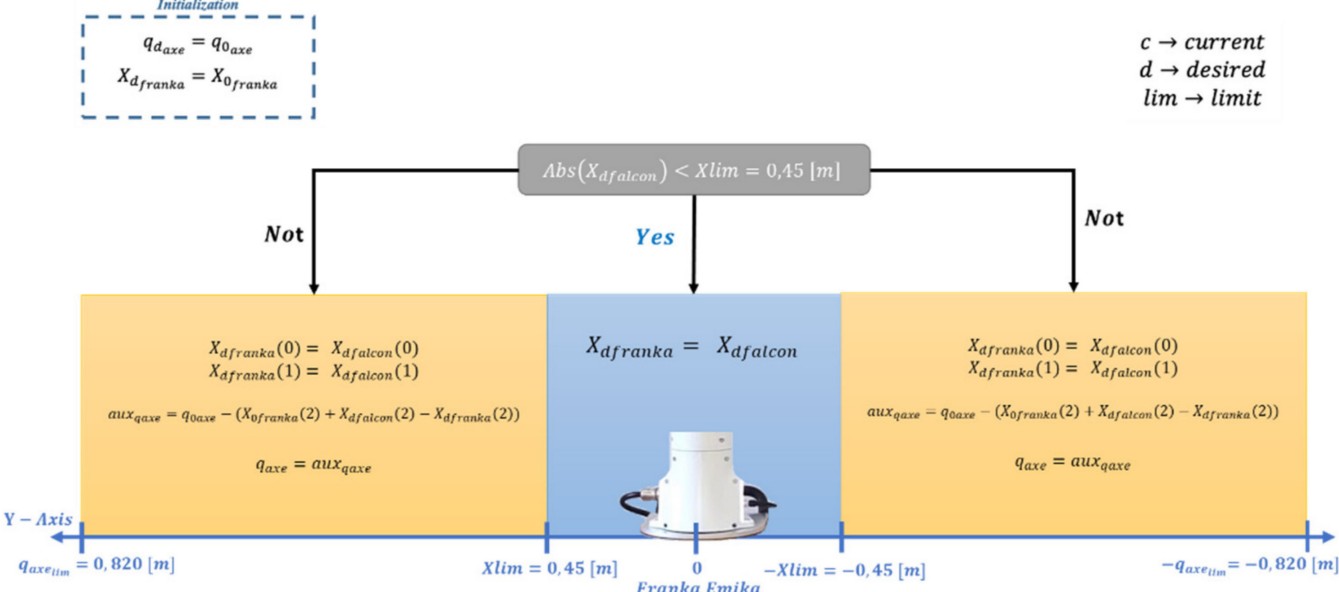

**Figure 13.** Movement distribution between the Franka Emika cobot and the linear axis in the *pos_criteria* node.

Evidently, the cobot always executes the movements along the remaining directions since the linear axis only can be actuated along the *Y-axis*.

We note that $\mp 0.820$ m corresponds to the length of the linear axis, where 0 m is the linear axis origin (middle point). The desired position transmitted by the haptic device is $q_{d_{axe}}$. Figure 14 depicts the proposed switching strategy.

### 4.2. Adaptive Joint-Limit Avoidance Strategy

A suitable way to exploit the redundancy of the $n = 7$-DoF cobot in the application of Doppler sonography is to enlarge its cartesian workspace, mainly rotational, allowing the ultrasound probe to execute large rotational movements. To achieve this, a suitable strategy consists of moving away the joint positions from the mechanical limits $\left[ \mathbf{q}_{min_i}, \mathbf{q}_{max_i} \right]$. Thereby, the null-space control approach can be defined as follows,

$$\boldsymbol{\tau}_{\aleph} = \aleph(\mathbf{q})\boldsymbol{\tau}_{JLA} = \aleph(\mathbf{q}) \left[ K_{JLA} \left( \frac{\partial \mathbf{w}_{JLA}(\mathbf{q})}{\partial \mathbf{q}} \right)^T - \mathbf{D}_{JLA}\dot{\mathbf{q}} \right] \tag{6}$$

The torque vector $\boldsymbol{\tau}_{JLA} \in \Re^n$ maximizes the objective function $\mathbf{w}_{JLA}(\mathbf{q}) \in \Re$ and is projected to the null-space of $J(q)$ through $\aleph(\mathbf{q})$ to guarantee a compatibility with the

cartesian task performed by the cobot. The second term of $\boldsymbol{\tau}_{JLA}$ allows to stabilize the internal motion of the robot. Let us define the objective function as,

$$\mathbf{w}_{JLA}(\mathbf{q}) = -\frac{1}{2n}\sum_{i=1}^{n}\left(\frac{\mathbf{q}_i - \mathbf{q}_{c_i}}{\mathbf{q}_{max_i} - \mathbf{q}_{min_i}}\right)^2 \tag{7}$$

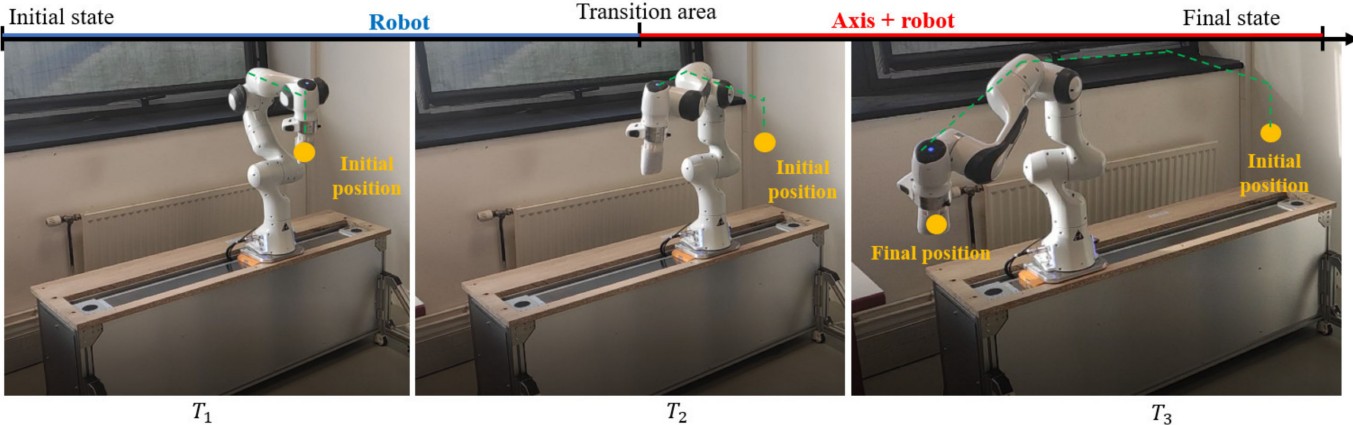

**Figure 14.** Switching robot/axis motion strategy. During a linear trajectory, the robot is initially activated to execute the desired motion. Once *Xlim* is reached (transition area), the axis is activated instead of the robot to execute the motion along the *Y-axis* (the robot keeps moving along the remaining axes).

The diagonal weighting matrix $\boldsymbol{K}_{JLA} \in \Re^{n \times n}$ enhances to ponderate each joint according to a prediction of those joints having higher risks to reach the mechanical limits during the execution of the cartesian task. Classical JLA strategies set $\boldsymbol{K}_{JLA}$ as constant, which is not an optimal way to exploit the redundant motion of the robot. First, if $\boldsymbol{K}_{JLA}$ is full-rank, it causes a permanent generation of null-space torques for all the joints that are not exactly at the middle point of the joint range $\mathbf{q}_{c_i}$, even if these joints are far of the mechanical limits. Moreover, the simultaneous avoidance of mechanical limits for different combination of joints can be incompatible with respect to the cartesian task and, therefore, insolvable by the robot.

In order to mitigate the limits of the classical approach, let us propose a new way to ponderate the weight of each joint when applying the JLA strategy of Equation (6). The goal is to adapt the weighting value of a joint *i* according to its proximity to the mechanical limits. Let's denote $\left(\mathbf{q}_{lim_{min_i}}, \mathbf{q}_{lim_{max_i}}\right)$ as the joint position thresholds indicating that the joint is approaching the mechanical limits. Between these two values the weighting value $\boldsymbol{K}_{JLA_i}$ is set to zero since it is estimated that the joint is far enough from the mechanical limits. Once one of the thresholds is reached, $\boldsymbol{K}_{JLA_i}$ linearly increases its value from the threshold and until the mechanical limit. Figure 15a compares the evolution of $\boldsymbol{K}_{JLA_i}$ for both, the classical JLA strategy and the proposed adaptive JLA strategy. Moreover, the consequences for the generation of the null-space torque term $\boldsymbol{K}_{JLA}\left(\frac{\partial \mathbf{w}_{JLA}(\mathbf{q})}{\partial \mathbf{q}}\right)^T$ are shown. When the proposed strategy is applied to the 7 joints, the proposed strategy generates zero null-space torques if all the joints move between the threshold limits (Figure 15b), which is an interesting advantage since it allows a manual reconfiguration of the elbow robot if needed. Moreover, the proposed strategy avoids the calculation of unneeded null-space torques and limits the case of incompatible torques acting in the nulls-space.

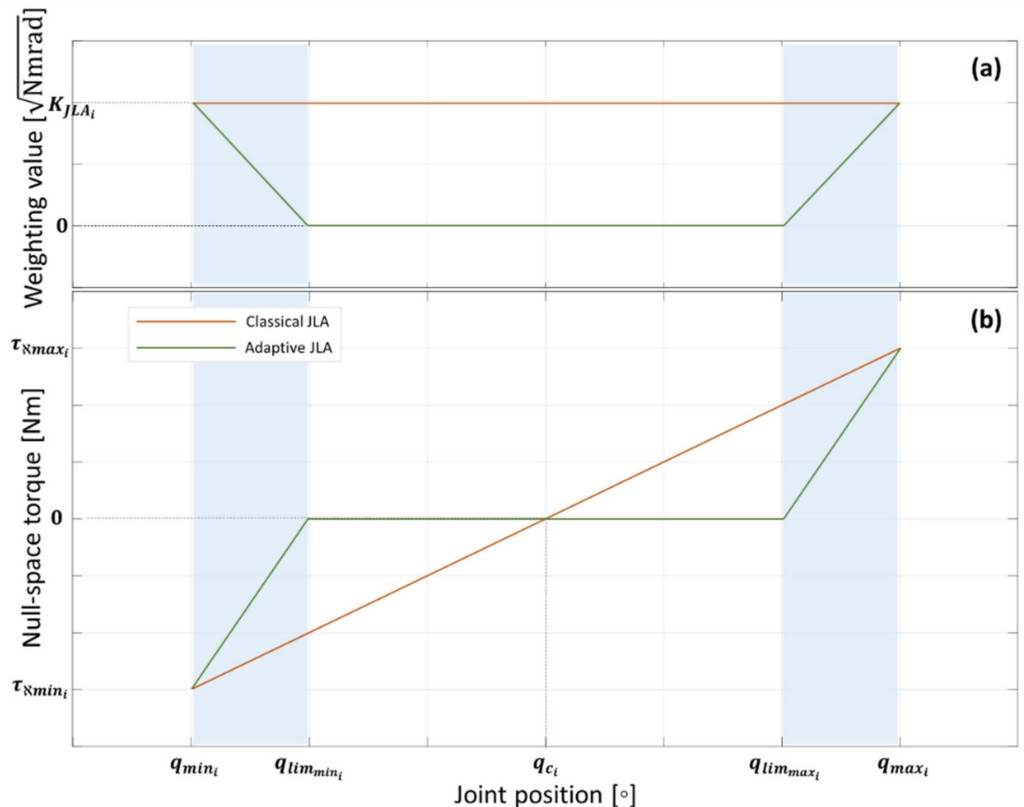

**Figure 15.** Evolution of (**a**) the weighting value and (**b**) the null-space torque for classical and adaptive JLA strategies along the position range of joint $i$.

### 4.3. Study Case

In order to validate the usefulness of the proposed adaptive joint-limit avoidance strategy, a situation typically encountered in a Doppler Ultrasound exam is investigated. Since the self-rotation axis of the ultrasound probe is coincident with the last robot joint, i.e., 7th joint, self-rotation movement only requests motions in the last robot joint. This assumption is correct if no null-space torques are applied to the control input. Therefore, the range of the self-rotation is limited to the one of the 7th joints, i.e., $(-166°, 166°)$ for the Franka Emika robot. This is frequently not sufficient to carry out common exams. The degree of redundancy of the anthropomorphic robot can then be used to enlarge the rotational workspace of the robot. Figure 16 shows the setup of the study case, where the robot holds the probe over the patient's body and a self-rotation motion is executed according to the orders of the expert site. For the sake of repeatability, the same desired self-rotation trajectory is applied for all the performed tests presented below.

First, a desired self-rotation motion is performed by the robot without a joint-limit avoidance strategy. Figure 17 compares the desired ($\phi_d$) and real ($\phi_r$) orientation trajectory, respectively. It is worth mentioning that the time delay evidenced between the two curves is due to the compliant behavior imposed by the control approach. It can be shown that the robot is unable to execute the desired trajectory since it reaches the mechanical limit of the 7th joint, as confirmed by the joint position trajectories (Figure 18). These results also confirm that only the 7th joint is requested to move in the absence of a null-space torque.

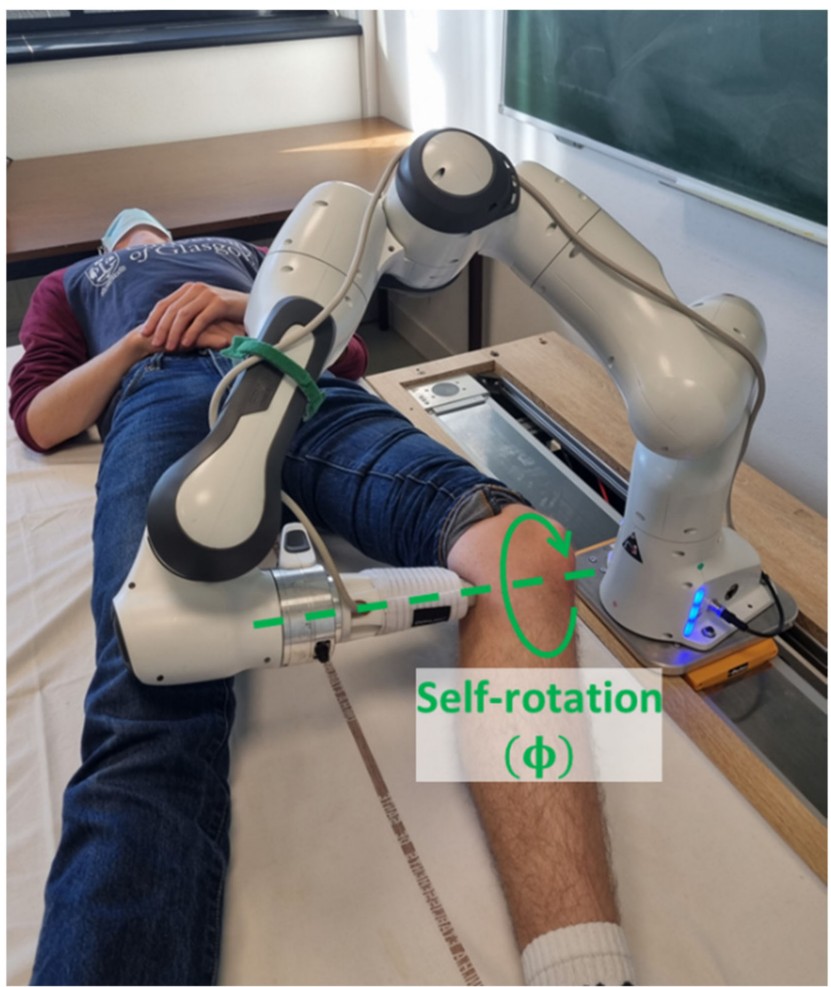

**Figure 16.** Exemplary image of the study case (no human subjects participated to the experiments). A self-rotation motion is imposed to the haptic probe during a Doppler ultrasound exam (see the accompanying Video S1).

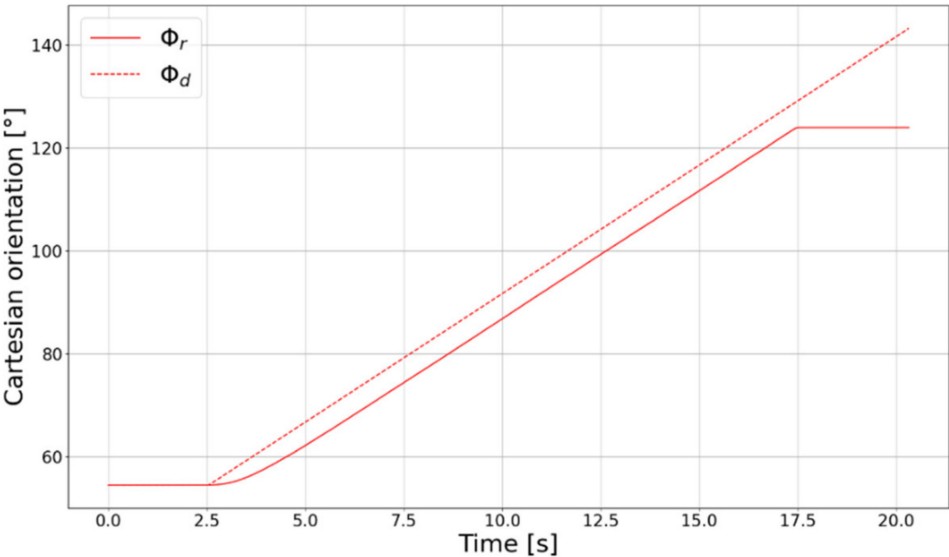

**Figure 17.** Comparison between the desired probe self-rotation $\phi_d$ and the executed self-rotation $\phi_r$ in the absence of a JLA strategy.

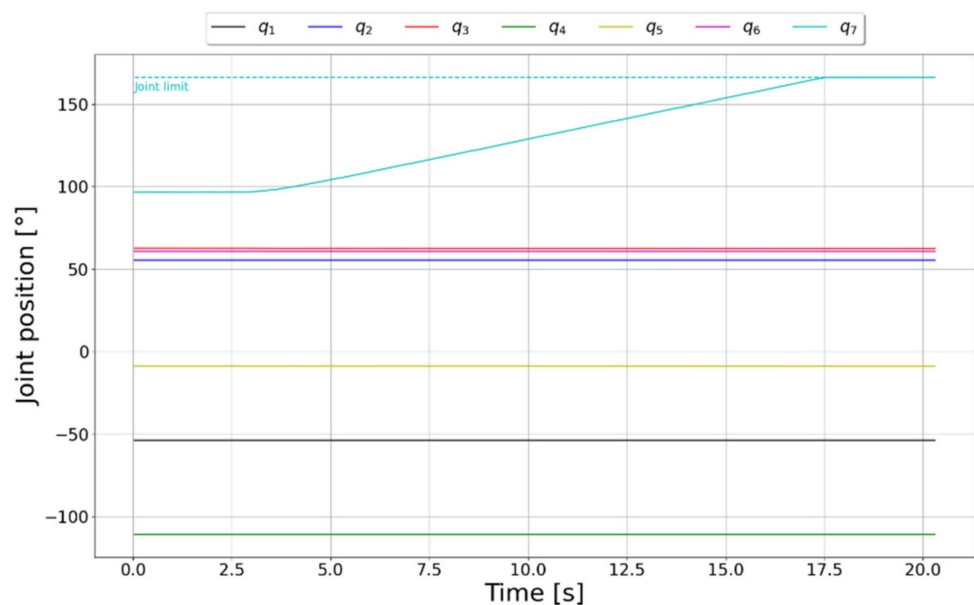

**Figure 18.** Joint position trajectories in the absence of a JLA strategy.

The same case is investigated when applying the joint-limit avoidance strategy of Equation (6). Initially, the classical JLA approach with constant values at the diagonal weighting matrix is evaluated. The performed experiences are resumed in Table 2 and the results are presented in Figure 19. Since we seek to avoid that the 7th joint reaches the mechanical limits, the first tests have been performed setting a non-zero constant value only to the 7th joint. Experiments have been performed with three different values ($K_{JLA_7} = 6$, 8 and 10 $\sqrt{Nmrad}$) and results in Figure 19 confirm that these values effectively enlarge the rotational robot's workspace moving away from the 7th joint from the mechanical limits. To do this, the robot's elbow motion, related to the degree of redundancy of the cobot, is reoriented accordingly. However, none of these tests allows completely following the desired trajectory since the elbow motion causes the 2nd joint to reach the mechanical limits. Therefore, three supplementary tests have been performed setting non-zero values to joints two and seven (see cases 4–6 in Table 2). Although the obtained results show that the rotational workspace can be enlarged depending on the combination of the constant values. This classical strategy is still limited because of the linear evolution of the null-space torques along with the joints' range, such as were explained in the previous section.

**Table 2.** Different weighting matrix choices for classical joint-limit avoidance strategy.

| Cases | $K_{JLA}\left(\sqrt{Nmrad}\right)$ |
|:-----:|:----------------------------------:|
| 1 | $diag([0, 0, 0, 0, 0, 0, 6])$ |
| 2 | $diag([0, 0, 0, 0, 0, 0, 8])$ |
| 3 | $diag([0, 0, 0, 0, 0, 0, 10])$ |
| 4 | $diag([0, 6, 0, 0, 0, 0, 6])$ |
| 5 | $diag([0, 8, 0, 0, 0, 0, 8])$ |
| 6 | $diag([0, 10, 0, 0, 0, 0, 10])$ |

Finally, we tested the proposed adaptive JLA strategy presented in Section 4.2. The control parameters of the strategy were tuned according to Table 3, where a margin of security of 30° has been set up between the joint limit thresholds and the mechanical limits provided by the constructor.

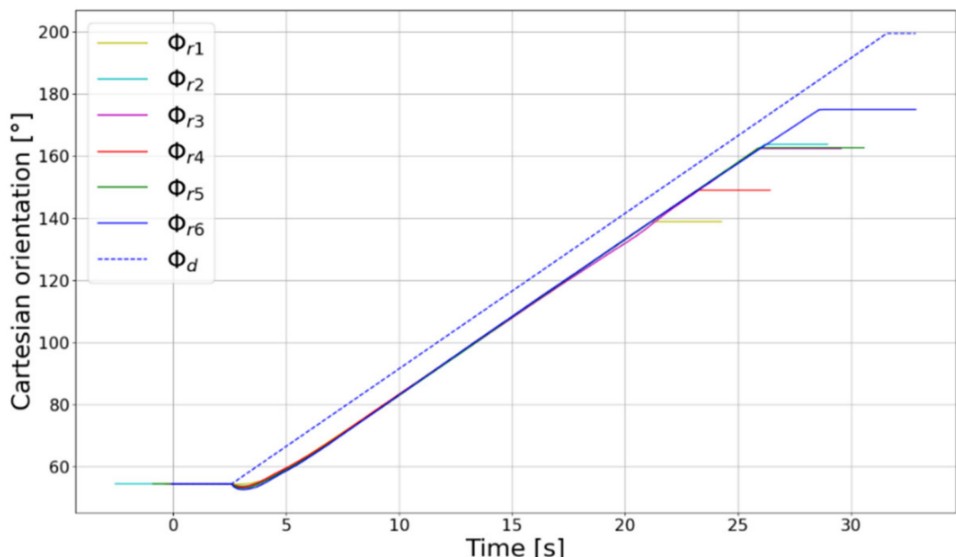

**Figure 19.** Comparison between the desired probe self-rotation $\phi_d$ and the executed self-rotation $\phi_r$ for the different constant weighting matrix choices presented in Table 1.

**Table 3.** Joint parameters applied for the adaptive JLA strategy. The mechanical limits $\mathbf{q}_{min}$ and $\mathbf{q}_{max}$ are provided by the constructor.

| | **Mechanical Limits** | | **Joint Limit Thresholds** | | **Weighting Limit** |
|---|---|---|---|---|---|
| **Joint** $i$ | $\mathbf{q}_{min}$ $(°)$ | $\mathbf{q}_{max}$ $(°)$ | $\mathbf{q}_{lim_{min}}$ $(°)$ | $\mathbf{q}_{lim_{max}}$ $(°)$ | $K_{JLA_i}$ $\left(\sqrt{Nmrad}\right)$ |
| 1 | −166 | 166 | −136 | 136 | 20 |
| 2 | −101 | 101 | −71 | 71 | 20 |
| 3 | −166 | 166 | −136 | 136 | 20 |
| 4 | −176 | −4 | −146 | −34 | 20 |
| 5 | −166 | 166 | −136 | 136 | 20 |
| 6 | −1 | 215 | 29 | 185 | 20 |
| 7 | −166 | 166 | −136 | 136 | 20 |

Figures 20–22 show the obtained results for the execution of the desired self-rotation trajectory. It can be seen in Figure 20 that the robot completely executes the desired trajectory due to the adaptive JLA strategy. It is worth mentioning that the offset observed at the end of the trajectory is caused by the tuning of low stiffness values in $\mathbf{K}_p$. Figures 21 and 22 also show that only joints two and seven reach the joint limit thresholds and, therefore, the other weighting joints rest at zero, avoiding the generation of useless null-space torques. Furthermore, unlike the classical JLA strategy, in this case, if none of the joints reaches the corresponding limit thresholds, no null-space torques are generated and the elbow robot can freely be moved by hand.

However, in order to endorse the correct functioning of the entire platform, a test phase in real conditions on multiple exams must be undertaken. This step will validate the conducted work and will initiate the improvement of the haptic interface to meet the angiologist's requirements. This first work opens the door to study multiple control strategies exploiting the degrees of redundancy of the platform.

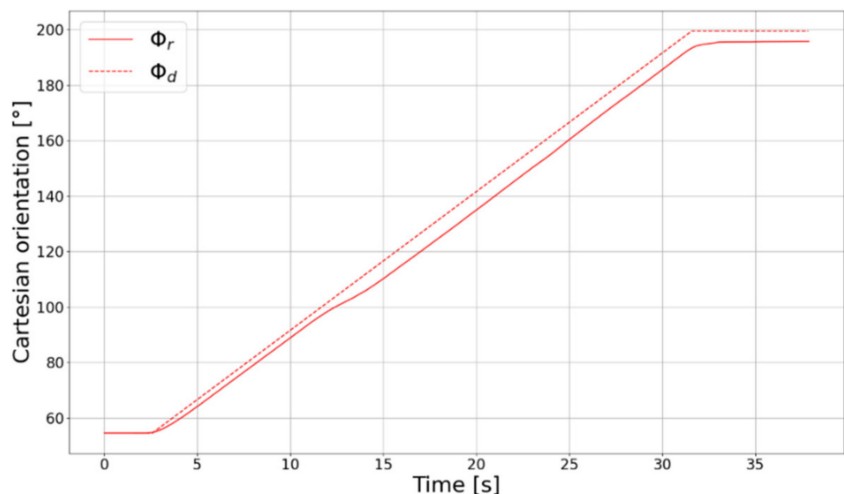

**Figure 20.** Comparison between the desired probe self-rotation $\phi_d$ and the executed self-rotation $\phi_r$ when applying the adaptive JLA strategy.

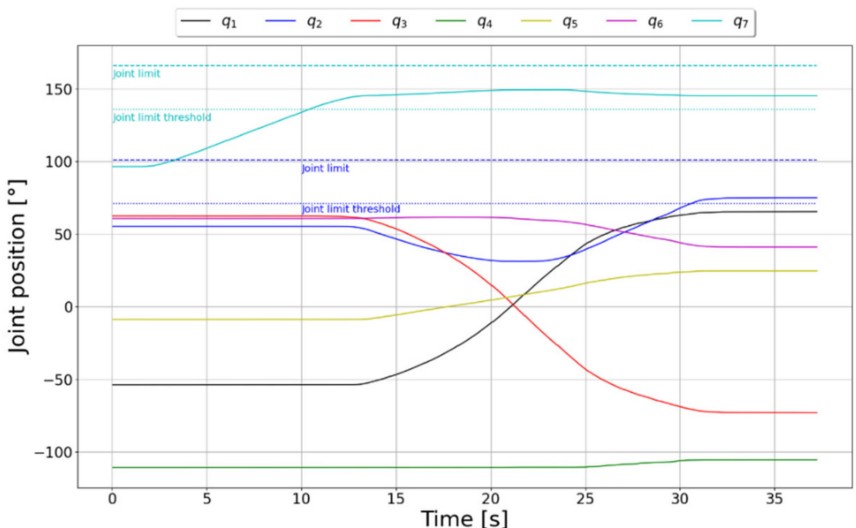

**Figure 21.** Joint position trajectories when applying the adaptive JLA strategy.

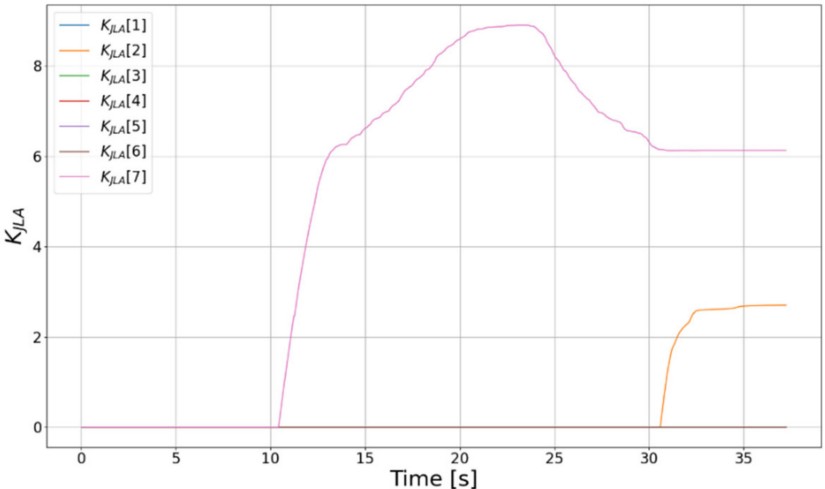

**Figure 22.** Evolution of the diagonal weighting values related to each robot joint for the adaptive JLA strategy.

## 5. Conclusions and Future Work

This paper presents an innovative redundant robotic assistant that combines a 7-DoF collaborative Franka robot and a linear axis to achieve a 7 + 1 DoF platform. This robot assistant is conceived to assist sonographers in Doppler Sonography exams and, unlike existing systems, is suitable for full body scanning without requiring manual repositioning of the robot base. The details of the kinematic design have been presented here, including the operator site (6-DoF hybrid haptic interface) and the patient site (7-DoF cobot + 1-DoF linear axis). Moreover, two alternative torque-based control laws have been proposed to deal with the degrees of redundancy of the platform, whereas a compliant behavior has been implemented for the execution of the main medical task, which is a valuable safety feature in case of undesired contacts with the patient. The first mode, called "fully redundant control mode", comprises the 8-DoF in the kinematic model so that the motion of the 8 axes is synchronized. Moreover, the prioritization of the linear axis motion is tuned by a null-space torque added to the torques performing the main task. The second control mode, "decoupled control mode", considers the two systems, the cobot and the linear axis, separately, and a switching strategy is proposed to launch the motion of the linear axis when the cobot is approaching the workspace limits. In addition, an adaptive JLA strategy has been proposed in this paper to exploit the redundancy of the cobot. In fact, unlike the classical strategies tuning a constant weighting matrix for the JLA strategy and generating unsuitable remaining torques, the new strategy proposes a variable weighting matrix, whose values are adapted according to the proximity of each robot joint to its mechanical limit. Therefore, the priority is judiciously given by order to the joints closest to their limits. In the opposite case, if none of the joints is near the mechanical limit, no additional null-space torques are generated. The application of this strategy naturally enlarges the workspace of the robot, as it has been shown for a study case presented for the examination of the inner side of a knee.

Future work will be dedicated to the medical validation of the experimental platform. It is planned to validate the usefulness of the proposed platform during Doppler sonography exams compared to the classical gesture, mostly in terms of time of execution and ergonomic conditions. Moreover, technical adjustments such as the velocity of motion of the linear axis and the choice of the optimal type of control will be studied. Finally, this work opens the door to the study of the proposed adaptive JLA strategy for several applications and robotic systems.

**Supplementary Materials:** The following supporting information can be downloaded at: https://www.mdpi.com/article/10.3390/act11020033/s1; Video S1: Caption of Figure 16 refers the accompanying video.

**Author Contributions:** E.G. and J.S. have designed the experiments and co-wrote the paper; A.T. has participated to the development of the platform; J.-M.G. is the Doppler sonographer who has validated the platform; the research work has been supervised by M.A.L., G.C. and S.Z. All authors have read and agreed to the published version of the manuscript.

**Funding:** This research was funded by the University of Poitiers and PPRIME Institute.

**Institutional Review Board Statement:** Not applicable.

**Informed Consent Statement:** Not applicable.

**Conflicts of Interest:** The authors declare no conflict of interest.

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
