# Peer review of "Redundancy Exploitation of an 8-DoF Robotic Assistant for Doppler Sonography"

_actuators, doi:10.3390/act11020033_

Round 1

Reviewer 1 Report

The paper is presenting a novel robotic platform for assisting ultrasound procedures, focusing on whole body Doppler examinations. The authors propose a novel teleoperated platform, composed on a custom designed operator device and 8DoF robotic system used on the patient. The authors are also describing two different approaches for exploiting the redundant DoFs on the patient site to optimize the scanning performance. The proposed system is experimentally evaluated both in simulation and with several real scenarios according to common methodologies and metrics.

The use of the terminology “master-slave” evokes strong feelings of degradation and human rights issues.  It carries with it the association to the brutal and dehumanizing institution of slavery.  Even though in the usage of the terms for tele-operated system there is no implied offense, the term harbors racist views.  I advise the authors to change the language of the article to make it more respectful, for instance “operator site – patient site” but there are many online resources providing list of alternatives. 

The manuscript is clearly written, even if several sections of the paper are missing important details for fully understanding the proposed method. The paper organization could be improved, for instance Section 5 is missing in the paper and Section 3 and 4 are mixing the description of the proposed control modes with their experimental evaluation. I suggest to separate the description of the proposed control modes from their experimental evaluation.

The novelty of the proposed work is doubtful, in particular because the introduction section is not able to correctly position this work with respect to previous works in literature. In the introduction section, the authors are not describing several important aspect of the proposed system, in particular:

  • Clinical motivations of the proposed system are not clearly described, neither the impact in the field. It is complex to objectively evaluate the proposed system considering the limited details provided in the introduction section. A better description of the clinical and technical motivations are essential, in particular the motivation of whole body doppler examination. According to my experience, whole body Ultrasound or Doppler examination are not very common, thus the authors should better clarify this point supporting their claim with bibliographic references.
  • The authors are not describing other robotic system for ultrasound procedure, highlighting the novel aspects of the proposed system. A more detailed description of other robotic system available in literature is required, in particular highlighting the improvement of the proposed robotic system with respect to other similar system.
  • The author are not correctly describing the novel contribution of this work with respect to previous works. In particular it is essential to clarify the novel contribution with respect to [7], [8], [9] regarding the robotic platform design. Moreover it is essential to clarify the novel contribution of the proposed control modes, since they seems to use quite standard approaches in exploiting the redundancy (as described for example in [18]).

In section 2, the authors are reporting results already presented in [7] in sub-section 2.1, while in section 2.2 many contents are already reported in [8]. I suggest the authors to remove contents that are already described in previous works, and focus on the novel contributions of this specific work, i.e., control modes to deal with system redundancy at the torque level.

I think also the abstract and the introduction should better describe the specific novel contribution of this work.

In the section 2.2.3 the authors are describing a platform based on ROS for real-time data exchange. The authors should clarify this point with more details, since standard ROS implementation is not guaranteeing any real-time capability.

The experimental evaluation described in section 3 and 4 is correctly performed, but the organization of these sections could be improved by separating the description of the proposed methods from their evaluation. Performing the evaluation of the two control modes in a similar manner (e.g. first in simulation and then in similar real experiments) could also help the reader in better understand and compare the performance of the two methods.

The discussions of the obtained results could be also improved, for instance with a better comparison with results obtained by relevant/close works available in literature.

The ethical protocol is not described in section 4.3, since in figure 16 the proposed system is showed while scanning a patient. The authors must describe the ethical protocol followed while performing these experiments and some details about its approval. To completely solve it I suggest to the authors the modification of Fig. 16 (for instance by showing the robotic system while scanning a mannequin or a leg synthetic phantom) and further clarify this point in the main text of the manuscript.

Author Response

Dear Reviewer, 

We appreciate the time and effort that you have dedicated to providing your valuable feedback on our manuscript. We are grateful to you for your insightful comments. We have been able to incorporate changes to reflect most of the suggestions provided by you and we have highlighted the changes within the manuscript. 

Here is a point-by-point response to your comments and concerns.

The paper is presenting a novel robotic platform for assisting ultrasound procedures, focusing on whole body Doppler examinations. The authors propose a novel teleoperated platform, composed on a custom designed operator device and 8DoF robotic system used on the patient. The authors are also describing two different approaches for exploiting the redundant DoFs on the patient site to optimize the scanning performance. The proposed system is experimentally evaluated both in simulation and with several real scenarios according to common methodologies and metrics.

The use of the terminology “master-slave” evokes strong feelings of degradation and human rights issues.  It carries with it the association to the brutal and dehumanizing institution of slavery.  Even though in the usage of the terms for tele-operated system there is no implied offense, the term harbors racist views.  I advise the authors to change the language of the article to make it more respectful, for instance “operator site – patient site” but there are many online resources providing list of alternatives. 

Response: Thank you for this valuable comment. The use of the terminology master-slave by the authors is based on the official terminology of the IFtoMM Federation (please refer to http://www.iftomm-terminology.antonkb.nl/2057/05.html). Nevertheless, we also consider that, in our case, an alternative to the master-slave terminology can be easily found to avoid any controversy. Therefore, we have opted to use “expert site-patient site” instead of “master-slave” in the revised paper version.

The manuscript is clearly written, even if several sections of the paper are missing important details for fully understanding the proposed method. The paper organization could be improved, for instance Section 5 is missing in the paper and Section 3 and 4 are mixing the description of the proposed control modes with their experimental evaluation. I suggest to separate the description of the proposed control modes from their experimental evaluation.

Response: Thank you for these comments. We have carefully modified the revised paper version to take them into account. In particular, we have at first modified the numbering of the last section (5 instead of 6), as requested. As for Sections 3 and 4 they present two different (alternative) control modes. The one presented in Section 3 is validated by simulation, whereas the control mode presented in Section 4 is validated experimentally. Since the objective of this work is not to compare the performance of the two control modes but to propose both alternatives to the public, we consider that the message may be noisy if we group the descriptions in one section and the validations in other section. We have added a paragraph before the beginning of Section 3 to allow readers to better understand the organization of sections 3 and 4 and clarify the contents given in each section as follows

“...Two alternative control modes for the platform at the patient site are proposed in the following sections. The first control mode, presented in section 3, considers the kinematics of the robot as a fully 8-DoF system. Therefore, the motion of the 8 axes are synchronized to fulfill the desired task and the two degrees of redundancy guarantee the avoidance of singularities of the robot arm. This control mode has for now been validated in simulation. The second control mode, presented in section 4 and validated experimentally, decouples the motion of the linear axis from the robot arm. The linear axis is only activated if the robot arm approaches its workspace limits. Furthermore, the remaining degree of redundancy of the robot arm is employed to avoid the joint limits through a new adaptive JLA strategy...” 

The novelty of the proposed work is doubtful, in particular because the introduction section is not able to correctly position this work with respect to previous works in literature. In the introduction section, the authors are not describing several important aspect of the proposed system, in particular:

  • Clinical motivations of the proposed system are not clearly described, neither the impact in the field. It is complex to objectively evaluate the proposed system considering the limited details provided in the introduction section. A better description of the clinical and technical motivations are essential, in particular the motivation of whole body doppler examination. According to my experience, whole body Ultrasound or Doppler examination are not very common, thus the authors should better clarify this point supporting their claim with bibliographic references.

Response: Clinical motivations are extensively detailed in section 2. In fact, section 2.1 “Medical requirements” has been written to clearly explain the clinical and technical motivations of the Doppler Ultrasound application. In order to complete our proposals, we have provided new elements to this section, including a new reference:

Evans K, Roll S, Baker J. Work-Related Musculoskeletal Disorders (WRMSD) Among Registered Diagnostic Medical Sonographers and Vascular Technologists: A Representative Sample. Journal of Diagnostic Medical Sonography 2009;25(6):287-299.

  • The authors are not describing other robotic system for ultrasound procedure, highlighting the novel aspects of the proposed system. A more detailed description of other robotic system available in literature is required, in particular highlighting the improvement of the proposed robotic system with respect to other similar system.

Response: Introduction section has been rearranged to improve the clearness of the proposed work. Additional references focus on the narrow area and further explanation have been added to illustrate the main contribution of this paper. For instance, the following text has been added in the revised paper

“...To address this issue, several teleoperated robotic solutions have been proposed in the recent years, mostly using commercial robotic arms as probe-holder [9-11]. We have also recently proposed a teleoperated robotic assistant using a 7-DoF anthropomorphic arm as probe-holder [7, 8]. The practitioner controls the robot by handling a haptic interface into a comfortable workspace. The main drawback of these proposed systems is the limited robot workspace that does not allow to perform an exam in the whole patient’s body. Mobile solutions, such as the one proposed in [12], overcome this problem but need the aid of a human assistant to hold the mobile robot over the patient. A new version of the system proposed in [8] has recently been introduced in [9]...” 

  • The author are not correctly describing the novel contribution of this work with respect to previous works. In particular it is essential to clarify the novel contribution with respect to [7], [8], [9] regarding the robotic platform design. Moreover it is essential to clarify the novel contribution of the proposed control modes, since they seems to use quite standard approaches in exploiting the redundancy (as described for example in [18]).

Response: We have added a paragraph resuming the contributions of the paper at the end of the Introduction of the revised paper as follows 

“... The main contributions of the paper are summarized as follows: a) detailed presentation of an 8-DoF teleoperated platform for Doppler sonography, b) validation of a fully redundant control mode at the torque level for the 8-DoF robotic system and c) introduction of a new adaptive JLA strategy for redundant robots that are controlled by torque via an optimal variation of the weighting matrix...” 

We have also modified the paper to better explain the differences between the proposed work and the ones presented in [7, 8] and [9].

Basically, [7, 8] propose the use of the 7-DoF Franka robot at the patient site, [9] introduces the mechanical design of the new 8-DoF system and the presented work details the communication framework as well as the proposed control modes.

In section 2, the authors are reporting results already presented in [7] in sub-section 2.1, while in section 2.2 many contents are already reported in [8]. I suggest the authors to remove contents that are already described in previous works and focus on the novel contributions of this specific work, i.e., control modes to deal with system redundancy at the torque level.

Response: Section 2 aims at showing the motivations of the proposed robotic platform, including the main results provided by the study presented in [7]. We consider that the description of the platform presented in section 2.2. is important for the reader, especially since more details have been provided here with respect to [9], including a specific communication framework.

I think also the abstract and the introduction should better describe the specific novel contribution of this work.

Response: As mentioned above, the introduction section has been substantially modified, among others, detailing the main contributions of the paper. The abstract has also been revised as requested.

In the section 2.2.3 the authors are describing a platform based on ROS for real-time data exchange. The authors should clarify this point with more details, since standard ROS implementation is not guaranteeing any real-time capability.

Response: The section 2.2.3 has been revised to emphasize the clearness of ROS implementation as requested. In particular, ROS runs real-time priority threads since a real-time kernel patch has been installed in Linux Ubuntu. In this way, a soft real-time operating system is configured.

The experimental evaluation described in section 3 and 4 is correctly performed, but the organization of these sections could be improved by separating the description of the proposed methods from their evaluation. Performing the evaluation of the two control modes in a similar manner (e.g. first in simulation and then in similar real experiments) could also help the reader in better understand and compare the performance of the two methods.

Response: As previously mentioned, Sections 3 and 4 present two alternative control modes, respectively. The one presented in Section 3 is validated by simulation, whereas the control mode presented in Section 4 is validated experimentally. Since the objective of this work is not to compare the performance of the two control modes but to propose both alternatives to the public, we consider that the message may be noisy if we group the descriptions in one section and the validations in other section. Nevertheless, we have added a paragraph before the beginning of Section 3 to allow readers to better understand the organization of sections 3 and 4.

The discussions of the obtained results could be also improved, for instance with a better comparison with results obtained by relevant/close works available in literature.

Response: The discussion of the obtained results has been revised and extended also by taking into account additional references, which have been included in the introduction section as also mentioned in previous responses to the reviewer’s comments.

The ethical protocol is not described in section 4.3, since in figure 16 the proposed system is showed while scanning a patient. The authors must describe the ethical protocol followed while performing these experiments and some details about its approval. To completely solve it I suggest to the authors the modification of Fig. 16 (for instance by showing the robotic system while scanning a mannequin or a leg synthetic phantom) and further clarify this point in the main text of the manuscript.

Response: Thank you for this remark. As described in the paper, the performed experimentations did not require any ethical approval, since it has been executed without involving any human. Fig. 16 describes the robot’s position to illustrate the study of a knee, but the experiments have been performed in free space (as shown in the accompanying video). For the sake of clarity, additional information has been added to the legend of the figure as follows

“Figure 16. Exemplary image of the study case (no human subjects participated to the experiments). A self-rotation motion is imposed to the haptic probe during a Doppler ultrasound exam.”

Reviewer 2 Report

In this paper, the authors consider an 8-DoF redundant robot as probe-holder in a teleoperated platform for Doppler sonography. The proposed robot is composed of a 7-DoF robotic arm that is mounted on a 1-DoF linear axis. This work details the design of the platform and proposes two main control modes to deal with its redundancy at the torque level. Simulations and experimental results are presented to verify the effectiveness of the proposed control modes. The topic is quite interesting and this paper is well-written. I have the following minor comments for the further improvement of the paper.

1) The authors are suggested to add one or two more keyword in the paper.
2) Some figures are unclear, such as Fig. 1 and Fig. 5. Please fix them with high resolution.
3) The authors are suggested to present a pseudocode or procedure for the proposed algorithm in the paper.
4) The simulation conditions and environment in ROS are suggested to be clarified. In addition, the codes of ROS project are also suggested to be unloaded for check in open source community, for example GitHut.
5) The authors are suggested to present more future potential works in the conclusion part.

The paper is good writing and presents technical contributions, which could be accepted after a revision.

Author Response

Dear Reviewer,

We appreciate the time and effort that you have dedicated to providing your valuable feedback on our manuscript. We are grateful to you for your insightful comments. We have been able to incorporate changes to reflect most of the suggestions provided by you and we have highlighted the changes within the manuscript.

Here is a point-by-point response to your comments and concerns.

In this paper, the authors consider an 8-DoF redundant robot as probe-holder in a teleoperated platform for Doppler sonography. The proposed robot is composed of a 7-DoF robotic arm that is mounted on a 1-DoF linear axis. This work details the design of the platform and proposes two main control modes to deal with its redundancy at the torque level. Simulations and experimental results are presented to verify the effectiveness of the proposed control modes. The topic is quite interesting, and this paper is well-written. I have the following minor comments for the further improvement of the paper.

  1. The authors are suggested to add one or two more keywords in the paper.

Response: Thank you for this comment. For sake of precision, the following additional keywords have been added to improve the clearness of the paper.

“torque-control” and “Doppler sonography”

  1. Some figures are unclear such as Fig. 1 and Fig. 5. Please fix them with high resolution.

Response: For sake of clarity, Fig. 1 and Fig. 5 have been upgraded with higher resolution. We hope these modifications will meet the expectations of the reviewer.

  1. The authors are suggested to present a pseudocode or procedure for the proposed algorithm in the paper.

Response: Thank you for this interesting remark. For sake of precision, section 3 and 4 have been revised and further details have been added to improve the clearness of the proposed control modes.

  1. The simulations conditions and environment in ROS are suggested to be clarified. In addition, the codes of ROS project are also suggested to be unloaded for check in opensource community, for example GitHub.

Response: Authors absolutely agree with the reviewer comment. For sake of precision, additional details have been added to the description of the ROS environment in the dedicated section. We hope these modifications will meet the expectations of the reviewer. Furthermore, we plan to create a GitHub repository in the following weeks to share our codes.

  1. The authors are suggested to present more future potential works in the conclusion part.

Response: As suggested, Conclusion and future work section has been extended. We hope these modifications will meet the expectations of the reviewer.

The paper is good writing and presents technical contributions, which could be acceptable after revision.

Round 2

Reviewer 1 Report

I would like to thank the authors for the time and efforts they have dedicated to reviewing this work, the manuscript has been edited following the reviewers' comments and the overall quality has improved significantly.

I have no further comments or questions for the authors